# Long-acting injectable antipsychotics for early psychosis: A comprehensive systematic review

Lulu Lian[1], David D. Kim[1,2], Ric M. Procyshyn[2,3], Diana Cázares[4], William G. Honer[2,3], Alasdair M. Barr[1,2]*

1 Department of Anesthesiology, Pharmacology & Therapeutics, University of British Columbia, Vancouver, B.C., Canada, 2 British Columbia Mental Health & Substance Use Services Research Institute, Vancouver, B.C., Canada, 3 Department of Psychiatry, University of British Columbia, Vancouver, B.C., Canada, 4 Department of Chemical & Biological Sciences, Universidad de las Americas Puebla, Puebla, Mexico

* al.barr@ubc.ca

**Data Availability Statement:** All relevant data are within the paper.

## Abstract

### Aim

Long acting injectable (LAI) antipsychotics are an alternative to oral antipsychotic (OAP) treatment and may be beneficial for patients in the early stages of schizophrenia. This study aims to provide a comprehensive review on the efficacy of first-generation and second-generation LAI antipsychotics in recent-onset, first-episode, and early psychosis patients.

### Methods

MEDLINE, EMBASE, PsycINFO, and Web of Science Core databases were used to search for studies that used LAIs in early psychosis patients. Studies published up to 06 Jun 2019 were included with no language restrictions applied. Inclusion criteria were a diagnosis of schizophrenia or related disorder, where patients were in their first episode or had a duration of illness ≤5 years.

### Results

33 studies were included: 8 RCTs, 4 post-hoc analyses, 2 case reports, and 19 naturalistic studies. The majority of studies evaluated risperidone LAIs (N = 14) and paliperidone palmitate (N = 10), while the remainder investigated fluphenazine decanoate (N = 3), flupentixol decanoate (N = 2), and aripiprazole (N = 1). Two studies did not specify the LAI formulation used, and one cohort study compared the efficacy of multiple different LAI formulations.

### Conclusions

While the majority of data is based on naturalistic studies investigating risperidone LAIs or paliperidone palmitate, LAIs may be an effective treatment for early psychosis patients in terms of adherence, relapse reduction, and symptom improvements. There is still a need to conduct more high quality RCTs that investigate the efficacy of different LAI formulations in early psychosis patients.

**Funding:** The funders had no role in study design, data collection and analysis, decision to publish, or preparation of the manuscript.

**Competing interests:** WGH has received consulting fees or sat on paid advisory boards for the Canadian Agency for Drugs and Technology in Health, AlphaSights, In Silico (unpaid), Newron, Translational Life Sciences and Otsuka/Lundbeck. RMP has received consulting fees or sat on paid advisory boards for Janssen, Lundbeck and Otsuka; is on the speaker's bureau for Janssen, Lundbeck and Otsuka. All other authors have no conflict of interest to declare.

# 1. Introduction

Antipsychotic medications are used as the primary pharmacological treatment for schizophrenia and associated psychotic disorders [1]. It has been shown that patients with a shorter duration of untreated psychosis experience better response to treatment with antipsychotics in terms of symptom and functional improvements [2, 3]. Therefore, it is important that treatment is initiated as soon as possible after the first episode of psychosis, which often occurs during late adolescence and early adulthood [1]. Nonadherence to medication is a major issue when attempting to treat schizophrenia [4–6]. Significant predictors of medication nonadherence include substance misuse, depressive symptoms, poor disease insight, and lower occupational status [4, 5]. Unfortunately, patients that are nonadherent are more likely to experience more involuntary readmissions to the hospital and frequent relapses [7]. Therefore, it is clear that medication nonadherence is a major issue that needs to be addressed when managing early psychosis. In addition to improving the clinical outcomes of patients, improving medication adherence also provides economic benefits. It has been estimated that the average cost for hospital readmission exceeds $8000 (US) per individual with schizophrenia [8].

Second-generation oral antipsychotic drugs (OAPs) are often suggested as a first-line treatment during the acute phase of early psychosis [9], and are widely used in youth for multiple indications [10–12]. Following control of the acute phase of psychosis, some have highlighted the benefits of switching patients to a depot antipsychotic [13, 14], also known as long-acting injectables (LAIs). The LAIs can be dosed less frequently than OAPs and are often used in patients with poor medication adherence, preventing negative clinical outcomes such as symptom relapse [15, 16]. In addition, a recent meta-analysis has found that LAIs were associated with reduced hospitalizations and emergency room admissions relative to OAPs in patients with schizophrenia, and importantly, these clinical benefits of LAIs were achieved while remaining cost-neutral relative to OAPs, where higher pharmacy costs associated with the use of LAIs were offset by lower medical costs [15].

A previous meta-analysis conducted on randomized controlled trials (RCTs) that evaluated the efficacy of LAIs versus OAPs for recent-onset psychosis patients found that LAIs were superior to OAPs in terms of adherence rates and discontinuation due to inefficacy [17]. However, there were no significant differences found between LAIs and OAPs in terms of relapse rates [17]. Previous systematic reviews have concluded that LAIs are an effective and safe treatment for first-episode and early psychosis patients [14, 18–21]. However, most of these reviews included studies that examined the use of risperidone LAI (RLAI) or only included RCTs. The present study aims to provide a more comprehensive review that includes RCTs, naturalistic studies, case studies, and post-hoc analyses that examine the use of both first-generation and second-generation LAIs in early psychosis patients, and therefore will offer results that are less biased and have increased generalizability and real-world applicability.

# 2. Materials and methods

## 2.1 Search strategies

We conducted a search for studies examining the efficacy of LAIs in early psychosis patients, including those that did not directly compare the efficacy of LAIs to OAPs. This systematic review was conducted according to the preferred reporting items for systematic reviews and meta-analyses guidelines (PRISMA) 2009 [22], as we have performed previously for systematic reviews on antipsychotic drugs [23, 24]. We searched for studies published from database inception using EMBASE (Ovid), MEDLINE (Ovid), PsycINFO (EBSCOhost), and Web of Science Core. The following MeSH terms or keywords were used in the search: 1) ['neuroleptic

agent MESH ' OR 'antipsychotic*' OR 'anti-psychotic*'] AND 2) ['depot' OR 'long-acting' OR 'long acting'] AND 3) '[first-episode psychosis OR first episode psychosis OR early psychosis OR early psychotic OR first psychotic episode OR recent-onset OR recent onset]. Additional studies were identified by searching the reference lists of relevant publications. No restrictions were placed on language, year, age, sex, ethnicity, setting, or trial duration. Two authors (LL, DC) independently screened the titles and abstracts of the studies and evaluated the full texts of the remaining eligible studies. Disagreements were discussed and resolved by agreement or discussion with another reviewer (DDK).

We included all studies that examined the use of LAIs in first-episode, recent-onset or early psychosis patients. We accepted studies that included patients with a mean duration of illness ≤5 years if there was no specified definition for recent-onset or early psychosis outlined in the study. To provide a comprehensive review, we considered all study designs except for conference papers for inclusion. Studies examining the same trial were included as long as they provided unique data from the associated trial.

## 2.2 Data extraction and outcome measures

Data were independently extracted by two authors (LL and DF). Information extracted from the studies included author information, year of publication, study design, trial duration, sample size, patient demographics (mean age, sex, ethnicity, diagnoses, duration of illness, comorbidities), antipsychotic used (including formulation), concomitant medications, medication dosages, and various outcomes of interest. We included a variety of outcomes including adverse events (AEs), discontinuation rates, relapse rates, adherence rates, symptom improvements, rehospitalization, and cognitive changes. We used the definitions for adherence and relapse as outlined by study authors.

# 3. Results

## 3.1 Search results and study characteristics

The literature search yielded 319 publications. After excluding duplicates, 252 articles remained (Fig 1). A total of 33 studies were selected to be included in the present systematic review. Table 1 summarizes the characteristics of the included studies. Of the included studies, two did not specify the LAI formulation used [25–27], 10 used paliperidone palmitate (PP) [27–36]. 14 used RLAIs [37–50] with 2 studies using the same trial [45, 47], 1 used aripiprazole LAI [51], and 5 used first-generation LAIs including fluphenazine decanoate [52–54] and flupentixol decanoate [55, 56]. One study investigated multiple LAI formulations including RLAI, olanzapine, PP, perphenazine, fluphenazine, flupentixol, aripiprazole, zuclopenthixol, and haloperidol [57].

## 3.2 Risperidone LAI (RLAI)

**3.2.1 Efficacy.** In the open-label RCT conducted by Malla et al. [44], RLAI and OAPs were found to be equally effective, with a mean total PANSS decrease of -18.1 (22.48); p<0.05 found in the RLAI group and a decrease of -17.7 (16.45); p<0.05 in the OAP group. Similarly, the open-label study conducted by Kim et al. [41] found no significant differences between groups in terms of PANSS total improvements (p = 0.907) and no statistically significant differences in CGI-S or PANSS scores were found at any timepoint between the RLAI and OAP groups in the open-label RCT conducted by Weiden et al. [50].

A post-hoc analysis of a 52-week randomized, double-blind study investigating the effects of 25 mg or 50 mg RLAI in patients with schizophrenia and schizoaffective disorder stratified

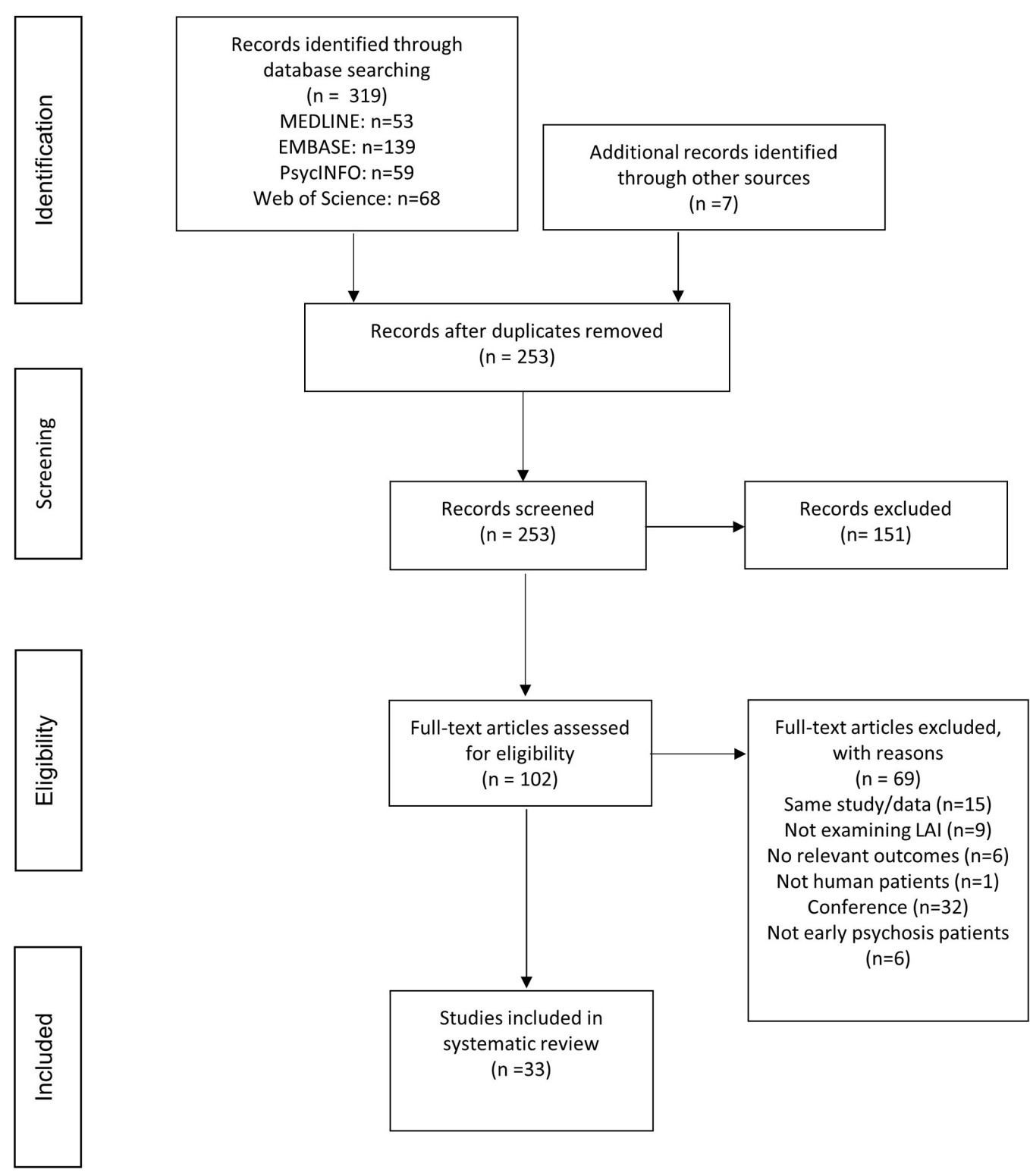

**Fig 1. PRISMA flow diagram.**

**Table 1. Summary of included studies.**

| Author, year | n | Study design | Duration (months) | Diagnosis | Patient population details | Mean Age (years) (SD) | Treatment | Outcomes of interest |
|---|---|---|---|---|---|---|---|---|
| Abdel-Baki 2019 | **LAI:** 17 **OAP:** 121 | Naturalistic, Prospective, Retrospective study | 36 | First-episode Schizophrenia (DSM-IV) with SUD (Drug Use Scale and Alcohol Use Scale) | Previous antipsychotic treatment: ≤ 1 year Comorbid SUD: n = 125 (89.9%) | **LAI:** 24.2 (3.2) **OAP:** 23.6 (3.8) | Not specified | Adherence, Discontinuation, Hospitalization, Relapse |
| Alphs 2018 | **LAI:** 42 **OAP:** 35 | Open-label RCT | 15 | Recent-onset Schizophrenia (Mini International Neuropsychiatric Interview, version 6.0) with history of criminal justice system involvement | Mean duration of illness: LAI: ~2.9 years OAP: ~3.2 years | **LAI:** 30.8 (9.7) **OAP:** 32.8 (10.9) | **LAI:** PP (78–234 mg) **OAP:** ARI, HAL, OLA, PAL, PER, QUE, RIS | Adverse events, Discontinuation, Hospitalization |
| Barrio 2013 | **LAI:** 26 **OAP:** 26 | Case-Control/ Naturalistic | 24 | Recent-onset Schizophrenia (DSM-IV) | Mean duration of illness: LAI: ~1.2 years OAP: ~0.4 years | **LAI:** 26.9 (6.7) **OAP:** 27.4 (7.5) | **LAI:** RIS (25–50 mg) **OAP:** OLA, RIS, CLO, ZIP, ARI, PAL, AMI, QUE | Symptom improvements, Hospitalization, Remission |
| Bartzokis 2011 | **LAI:** 11 **OAP:** 13 **CONTROL:** 13 | RCT | 6 | First-episode Schizophrenia or Schizoaffective disorder (DSM-IV) | Mean duration of illness: LAI: ~0.5 years OAP: ~0.5 years | **LAI:** 25.4 (4.8) **OAP:** 23.5 (4.6) | **LAI:** RIS (12.5–37.5 mg) **OAP:** RIS 1–7.5 mg | Cognition, White matter volume change |
| Bossie 2017 | **Recent onset** (≤5 years): 206 **Chronic** (> 5 years): 461 | Multi-phase study: Open-label phases and double-blind relapse prevention phase | 3 month OL acute treatment, 2.75 month OL stabilization, 15-month DB relapse prevention | Schizoaffective disorder (DSM-IV) experiencing a recent exacerbation of symptoms | Mean duration of illness: Recent onset: 2.9 (1.4) years Chronic: 18.1 (9.78) years | 33.8 (10.2) | **LAI:** PP 78–234 mg | Adverse events, Relapse, Discontinuation, Remission, Symptom improvements, |
| Cervone 2015 | 7 | Retrospective Study/ Naturalistic | 6 | First-Episode Psychosis (Affective and Non-affective) (DSM-IV-TR) | Mean duration untreated psychosis: 1 month to 10 years | 33.5 (11.5) | **LAI:** PP 75–150 mg | Adverse events, Discontinuation, Hospitalization, Relapse, Symptom improvements, |
| Chiliza 2015 | 207 | Retrospective study/ Naturalistic | 12 | Schiozphrenia, Schizophreniform, Schizoaffective (DSM-IV) | Mean duration untreated psychosis: 78.38 (147.65) weeks | 25.9 (6.9) | **LAI:** FLU 5–30 mg | Adverse events, Discontinuation, Relapse, Remission, Symptom Improvements |
| Dubois 2014 | **Electronic Schizophrenia Treatment Adherence Registry (e-STAR): 155 Trial for the Initiation and Maintenance of Remission in Schizophrenia with risperidone TIMORES: 105** | Post-hoc analysis | **e-STAR:** 24 **TIMORES:** 12 | Schizophrenia | Mean duration of illness: e-STAR: 1.2 (1.2) years TIMORES: 3 (3.9) years | **e-STAR:** 34.5 (13.8) **TIMORES:** 30.8 (7.0) | **LAI:** Risperidone (25–75 mg) | Discontinuation, Hospitalization, Remission, Symptom Improvements |

*(Continued)*

**Table 1.** (Continued)

| Author, year | n | Study design | Duration (months) | Diagnosis | Patient population details | Mean Age (years) (SD) | Treatment | Outcomes of interest |
|---|---|---|---|---|---|---|---|---|
| Emsley 2008 | **LAI:** 50 **OAP:** 47 | Post-hoc analysis LAI study: Open-label OAP: Double-blind RCT | 24 | Early Schizophrenia, Schizophreniform disorder, Schizoaffective disorder (DSM-IV) | Mean duration of illness: $\leq$ 1 year and $\leq$ 2 psychiatric hospitalizations Previous antipsychotic treatment: $\leq$ 12 weeks | **LAI:** 25.4 (7.4) **OAP:** 25.9 (5.8) | **LAI:** RIS (25–50 mg) **OAP:** RIS, HAL | Adverse events, Discontinuation, Relapse, Remission, Symptom improvements |
| Fàbrega 2015 | 2 | Case report | Not specified | **Subject A:** Undifferentiated Schizophrenia (DSM-IV) **Subject B:** Psychotic Disorder NOS (DSM-IV) | **Subject A:** Admitted to hospital following 2 years of total seclusion **Subject B:** Onset of symptoms 2 years prior to admission | **Subject A:** 14 **Subject B:** 17 Total: 15.5 (1.5) | **Subject A:** LAI: PP (50 mg/28 day) ORAL: ARI (5 mg/ day), PAL (6 mg/ day) **Subject B:** LAI: PP (50 mg/28 day), ZUC (100 mg/14 day) ORAL: ARI (15 mg/day), PAL (3 mg/ day) | Adverse events, Adherence, Hospitalization, Symptom improvements |
| Giordano 2020 | 50 | Exploratory Study | 12 | Schizophrenia (DSM-V) | First-episode psychosis inpatients Age-at-onset: 18–21 years: n = 14 (28%) 22–26 years: n = 36 (72%) Comorbid SUD: n = 32 (64%) | 23.6 (2.8) | **LAI:** ARI (400 mg) | Adherence, Adverse events, Discontinuation, Efficacy |
| Kane 1982 | **Treatment:** 11 **Placebo:** 17 | Double-blind RCT | 12 | Schizophrenia, Unspecified functional psychosis, other psychiatric disorder, manic disorder with schizotypal features, major depressive disorder with schizotypal features (Research Diagnostic Criteria) | All patients had only 1 previous schizophrenic episode with stable remission for at least 4 weeks and to a maximum of 1 year following hospital admission Mean length of remission: 16.9 (7.0) weeks | 21.9 (4.3) | **LAI:** FPZ (12.5–50 mg) ORAL: FPZ HCl | Adverse events, Relapse, Remission |
| Kim 2008 | **LAI:** 22 **OAP:** 28 | Naturalistic, controlled Open-label study | 24 | First-episode Schizophrenia or Schizoaffective disorder (DSM-IV (SCID)) | Mean duration of illness: **LAI:** 1.5 (1.5) years **OAP:** 2.2 (3.1) years | **LAI:** 32.5 (10.6) **OAP:** 31.0 (10.1) | **LAI:** RIS (Mean ± SD = 28.98 ± 6.00 mg) **OAP:** RIS (2.79 ± 0.92 mg) | Adherence, Relapse |
| Lasser 2007 | 66 | Open-label study/ Naturalistic | 11.5 | Schizophrenia, Schizoaffective disorder (DSM-IV) | Prior treatment duration: 131 (164.7) days | 23.3 (3.3) | **LAI:** RIS (25–50 mg) | Adherence, Discontinuation, Remission, Symptom improvements |

(*Continued*)

**Table 1.** (Continued)

| Author, year | n | Study design | Duration (months) | Diagnosis | Patient population details | Mean Age (years) (SD) | Treatment | Outcomes of interest |
|---|---|---|---|---|---|---|---|---|
| Mac-fadden 2010 | **Recent onset:** (≤3 years): 57 **Chronic:** (>3 years): 266 | Post-hoc analysis of an RCT | 12 | Schizophrenia, Schizoaffective (DSM-IV) | Mean duration of illness: **Recent-onset:** 1.9 (0.9) years **Chronic:** 18.4 (10.0) years | **Recent-onset:** 30.9 (11.4) **Later-onset:** 43.1 (10.9) | RLAI (25–50 mg) | Adverse events, Discontinuation, Relapse, Symptom improvements |
| Malla 2016 | **LAI:** 42 **OAP:** 35 | Open-label RCT | 24 | Recent-onset Schizophrenia, Schizophreniform, or Schizoaffective (DSM-IV (SCID)) | Mean duration of illness: Total sample: ~ 9 (0.88) months | **LAI:** 22.5 (3.1) **OAP:** 23.0 (2.9) | **LAI:** RIS (25–50 mg) **OAP:** OLA, QUE, RIS | Adherence, Adverse events, Discontinuation, Hospitalization, Relapse, Symptom improvements |
| Morken 2008 | **LAI:** 12 **OAP:** 38 | Case control/ Naturalistic | 24 | Schizophrenia, Schizoaffective disorder (DSM-IV) | Mean duration of illness: Less than 2 years | 25.4 (4.6) | **LAI:** NS | Adherence, Relapse, Hospitalization, Symptom improvements, |
| Naprye-yenko 2010 | 294 | Open-label study/ Naturalistic | 6 | Schizophrenia, Schizoaffective (DSM-IV) | Mean number of previous psychotic episodes: 2.4 (0.7) | Median age: female: 31 male: 27 | **LAI:** RIS (25–50 mg) | Adherence, Adverse events Discontinuation, Relapse, Remission, Symptom improvements |
| Olivier 2015 | 92 | Case control/ Naturalistic | 12 | Schizophrenia, Schizophreniform, Schizoaffective (DSM-IV) | Mean duration untreated psychosis: 31.86 (35.29) weeks | 24 (6.0) | **LAI:** FLU (10 mg) | Cognition, Discontinuation, Symptom improvements |
| Parellada 2005 | 382 | Post hoc subgroup analysis | 6 | Schizophrenia, Schizoaffective (DSM-IV) | Mean duration of illness: 1.5 (1.1) years | 29.0 (4.7) | **LAI:** RIS (25–50 mg) | Adherence, Adverse events, Discontinuation, Hospitalization, Relapse, Symptom improvements |
| Privat 2015 | **LAI:** 11 **OAP:** 177 | Naturalistic study | 6 | First-episode Schizophrenia, Schizophreniform Disorder, Brief Psychotic Disorder (DSM-V) | Mean duration untreated psychosis: LAI: 70.1 (65.4) days OAP: 109.5 (199.8) days | **LAI:** 22.2 (3.6) **OAP:** 24.9 (5.0) | **LAI:** PP, ZUC, RIS **OAP:** OLA, ARI, PAL, AMI, QUE, CLO, RIS | Hospitalization |
| Rabino-witz 2011 | 294 | Open-label study/ Naturalistic | 6 | Schizophrenia, Schizoaffective (DSM-IV) | Mean number of previous psychotic episodes: 2.4 (0.7) | Median age: Female: 31 Male: 27 | **LAI:** RIS (25–50 mg) | Symptom improvements in relation to PAS global assessment of highest level of functioning |

(*Continued*)

**Table 1.** (*Continued*)

| Author, year | n | Study design | Duration (months) | Diagnosis | Patient population details | Mean Age (years) (SD) | Treatment | Outcomes of interest |
|---|---|---|---|---|---|---|---|---|
| Rifkin 1977 | **LAI:** 23 **OAP:** 28 | Double-blind RCT | 12 | Schizophrenia—any subtype (diagnosis based on study psychiatrist using criteria outlined in Klein DF, Davis JM: Diagnosis and Drug Treatment of Psychiatric Disorders. Baltimore, Williams & Wilkins, 1969.) | Mean number of psychotic episodes: Acute patients: < 1 episode Chronic patients: LAI: 1.67 episodes OAP: 1.90 episodes | **LAI:** 23.6 (Range: 17–38) **OAP:** 23.8 (Range: 17–37) | **LAI:** FPZ (0.5–2.0 mL) **OAP:** FPZ (5–20 mg) | Adverse events, Discontinuation, Relapse |
| Ruan 2010 | 31 | Open-label study/ Naturalistic | 5.5 | Schizophrenia (DSM-IV-TR) | Not specified | 15.9 (3.3) | **LAI:** RIS (25–37.5 mg) | Adverse events, Discontinuation, Symptom improvements, |
| Schreiner 2015 | **LAI:** 352 **OAP:** 363 | Single-blinded RCT | 24 | Recent-onset Schizophrenia (DSM-IV) | Mean duration of illness: LAI: 3.0 (1.7) years OAP: 2.9 (1.5) years | **LAI:** 32.6 (10.7) **OAP:** 32.6 (10.1) | **LAI:** PP (25–150 mg) **OAP:** ARI, HAL, OLA, PAL, QUE, RIS | Adherence, Discontinuation, Relapse, Symptom improvements |
| Sliwa 2012 | **Recent- onset** (≤5 years): 216 **Chronic** (>5 years): 429 | Post-hoc analysis of a multiphase trial | 12 | Schizophrenia (DSM-IV) | Mean duration of illness: Recent onset: 2.9 (1.5) Chronic: 16.2 (8.1) | Recent-onset: 31.0 (9.3) Chronic: 40.6 (9.7) | PP (39–156 mg) | Adverse events, Discontinuation |
| Subotnik 2015 | **LAI:** 40 **OAP:** 43 | Open-label RCT | 12 | Recent-onset Schizophrenia, Schizoaffective disorder, Depressed type, Schizophreniform disorder (DSM-IV) | Mean duration of illness: LAI: 6.9 (6.8) months OAP: 7.9 (6.6) months | **LAI:** 21.9 (3.8) **OAP:** 21.1 (3.2) | **LAI:** RIS (12–37.5 mg) **OAP:** RIS (1.0–7.5 mg) | Adherence, Adverse events, Discontinuation, Hospitalization, Relapse |
| Taipale 2018 | 8719 | Cohort study/ Naturalistic | 240 | First-episode Schizophrenia (ICD-10, ICD-9, ICD-8) | First hospitalization patients Previous antipsychotic treatment: No use of antipsychotics for one year preceding first hospitalization | **Total:** Median: 36.2 (Range: 26.2–52.3) | **LAI:** OLA, PP, PER, FPZ, ARI, FLU, ZUC, RIS, HAL **OAP:** CLO, CPX, FLU, OLA, ZUC, LEV, ARI, RIS, HAL, PER, FPZ, THOR, QUE | Hospitalization |
| Titus-Lay 2018 | **LAI:** 4 **OAP:** 35 | Retrospective study/ Naturalistic | 12 | Recent-onset Schizophrenia, Schizophreniform disorder, Schizoaffective disorder, or Psychosis NOS (DSM-IV) | Mean duration of illness: ≤ 2 years | **Total:** 21 (NS) | **LAI:** PP **OAP:** ARI, OLA, HAL, FPZ, RIS | Adherence |

(*Continued*)

**Table 1.** (Continued)

| Author, year | n | Study design | Duration (months) | Diagnosis | Patient population details | Mean Age (years) (SD) | Treatment | Outcomes of interest |
|---|---|---|---|---|---|---|---|---|
| Vázquez-Bourg-non 2014 | 1 | Case report | Not specified | Paranoid Schizophrenia | Less than 2 years | 34 | **LAI:** PP (Initial: 100 mg/month Final: 75 mg/month with 200 mg sertraline) | Adverse events |
| Weiden 2012 | **LAI:** 19 **OAP:** 18 | Open-label RCT | 24 | First-Episode Schizophrenia, Schizophreniform, or Schizoaffective (DSM-IV (SCID)) | NS | **Total (LAI + OAP):** 25.3 (6.6) | **LAI:** RIS (25–50 mg) **OAP:** ARI, OLA, QUE, ZIP, RIS | Adherence, Adverse events Discontinuation, Hospitalization, Symptom improvements |
| Yee 1998 | **Recent onset** (≤2 years): 22 **Control:** 11 | Comparative controlled study/ Naturalistic | 3 | Schizophrenia, Schizophreniform, Schizoaffective (DSM-IV) | Mean duration of illness: Less than 2 years | 25.8 (5.4) | **LAI:** FPZ (10–15) **OAP:** RIS (2–9 mg) | Neurocognition: auditory P50 component of the event-related potential |
| Zhang 2015 | 521 | Open-label study/ Naturalistic | 18 | Schizophrenia (DSM-IV) | Mean duration of illness: Less than 5 years 36.7% less than 1 year | 28.7 (7.95) | **LAI:** PP (50–150 mg) | Adverse events, Hospitalization, Discontinuation, Symptom improvements, |

Abbreviations: DSM, Diagnostic and Statistical Manual; ICD, International Statistical Classification of Diseases and related Health Problems; NS, not specified; NOS, not otherwise specified; OL, open-label; RCT, Randomized controlled trial; SCID, Structured Clinical Interview for DSM; SUD, Substance use disorder; AMI, amisulpride; ARI, aripiprazole; CLO, clozapine; CPX, chlorprothixene; FLU, flupentixol; FPZ, fluphenazine; HAL, haloperidol; LEV, levomepromazine; OLA, olanzapine; PAL, paliperidone; PP, paliperidone palmitate; PER, perphenazine; QUE, quetiapine; RIS, risperidone; THOR, thioridazine; ZIP, ziprasidone; ZUC, zuclopenthixol.

data using time since diagnosis (≤ 3 years and > 3 years) [43]. The study demonstrated that recently diagnosed participants (≤ 3 years) experienced a significantly greater reduction in mean total PANSS score from baseline to last-observation-carried-forward (LOCF) end point (14.2% reduction; 64.8 (14.1) to 55.6 (16.4)), compared to those with > 3 years duration of illness (6.1% reduction; 66.8 (16.9) to 62.7 (17.0)) (p = 0.004) [43]. The case-control study conducted by Barrio et al. [37] also found that LAIs were superior to OAPs in terms of efficacy, with a mean total PANSS reduction of 40.3% in the RLAI group (79.9 (28.6) to 47.7 (12.0)) and 25.2% in the OAP group (88.5 (16.1) to 66.2 (18.5)) (p < 0.001).

In the post-hoc analysis conducted by Dubois et al. [39] that included two observational studies: e-STAR study (with 97 patients treated with RLAI and with a mean duration of illness of 1.2 years) and TIMORES study (with 83 patients treated with RLAI and with a mean duration of illness of 3 years), there was a significant 1-year mean CGI-S change from baseline of -1.19 (-1.472; -0.908); p<0.0001 in the e-STAR study and -1.40 (-1.71; -1.14); p<0.0001 in the TIMORES study. Also, the mean CGI-S change was significantly greater in patients in their early stages than those in their late stages of illness.

In the post-hoc analysis of two separate studies conducted by Emsley et al. [40], a significantly greater reduction in total PANSS score was observed from baseline to endpoint (24 months) in the RLAI compared to OAP group (-39.7 (21.1) versus -25.7 (30.2) p = 0.009. In the open-label study conducted by Lasser et al. [42], a reduction of 14.9% in mean total PANSS score from baseline to endpoint was observed (-9.7 (1.7); p <0.001) and 64% of participants had a reduction of ≥ 20% in their total PANSS score. Napryeyenko et al. [45] found a 32.7% reduction in total PANSS score (-13.0 (14.0); p <0.01) and 68.4% of patients obtained a total

PANSS score reduction of at least 20%. Rabinowitz et al. [47] examined the same trial included in the study by Napryeyenko et al. [45] and found that improvements in total PANSS score was greater in groups with stable-good premorbid functioning compared to stable-poor and deteriorating groups.

Parellada et al. [46] conducted an open-label study and found a 15.7% reduction in total PANSS score from baseline to endpoint (-11.3 (19.1); p<0.0001) and 40% of patients experienced an improvement in total PANSS score of ≥ 20%. Similarly, Ruan et al. [48] found a significant improvement in total PANSS score as a result of RLAI treatment (7.6% reduction; -4.4 ± 0.2; p<0.001) and 67.7% of patients achieved at least a 20% reduction in total PANSS total score. It may be helpful to note that the Lasser et al., Napryeyenko et al., Rabinowitz et al. and Parellada et al. studies did not include comparator groups.

**3.2.2. Adherence.** In the open-label RCT conducted by Malla et al. [44], there were 4 noncompliant subjects in the LAI group (90.5% adherent) and 9 non-compliant subjects (74.3% adherent) in the OAP group. In the open-label RCT conducted by Subotnik et al. [49], adherence was evaluated on a scale of 1–5, with 1 being perfect adherence and 5 being complete nonadherence. The mean adherence over the follow-up period on this scale was better in the LAI group (1.1 (0.5)) compared to the OAP group (1.9 (0.8)) (p < 0.001). Furthermore, 95% of patients in the RLAI group had an excellent adherence level (score <1.5) compared to 33% in the OAP group [50]. On the other hand, the open-label RCT conducted by Weiden et al. [50] did not find a significant difference in adherence levels for the LAI group (21% adherent) compared to the OAP group (17% adherent). In this study, non-adherence behaviour was defined as a period of ≥ 14 consecutive days of complete discontinuation of antipsychotic treatment.

The open-label study by Kim et al. [41] found that 68% of patients taking LAI had good adherence compared to 32% taking OAP, where adherence was defined by the number of actual visits to the outpatient clinic divided by the number of days that the patient was scheduled to visit the clinic. The open-label studies conducted by Parellada et al. [46], Lasser et al. [42] and Ruan et al. [48] all found that 3% of patients discontinued the study due to noncompliance. In the open-label study by Napryeyenko et al. [45], only 1% of patients discontinued the study due to noncompliance.

**3.2.3. Relapse.** In the open-label RCT conducted by Malla et al. [44] there was a greater relapse rate in patients taking RLAI (26%) compared to those taking OAPs (14.3%), whereas Subotnik et al. [49] found a relapse rate of 5% in patients taking RLAI and 33% in those taking OAPs (p<0.001). In the post-hoc analysis conducted by Macfadden et al. [43], 10.5% of patients recently diagnosed with schizophrenia relapsed compared to 21.8% of those that were diagnosed >3 years ago (chronic group). Patients in the chronic group were twice as likely to relapse compared to the recently diagnosed group at all time points throughout the study (HR: 2.2 (95% CI: 0.95, 5.13); p = 0.056).

Kim et al. [41] found a 2-year relapse rate of 23% in the LAI group and 75% in the OAP group whereas 3% of patients taking RLAI experienced a relapse in the study by Parellada et al. [46]. Emsley et al. [58] found a significantly lower relapse rate in the RLAI group (9.3%) compared to OAP group (42.1%) (p = 0.001). In the study by Napryeyenko et al. [45], a relapse rate of 2.5% was found among patients that experienced at least a 20% reduction on their total PANSS score during the study.

**3.2.4 Rehospitalization.** In the open-label RCT conducted by Malla et al. [44], the rehospitalization rate was 19% in the group taking RLAI compared to 11.4% in the OAP group. In the OL RCT by Subotnik et al. [50], there was a lower rehospitalization rate in the RLAI group (5%) compared to the OAP group (19%) (p = 0.05) [49]. The Kaplan-Meier rehospitalization estimate in the OL RCT conducted by Weiden et al. [50] was 26% for the RLAI group and 23%

for the OAP group at 52 weeks. Macfadden et al. [43] found that 3.5% of patients in the recently diagnosed group were hospitalized compared to 10.5% of those in the chronic group.

In the open-label study by Parellada et al. [46] 5% of patients were newly hospitalized throughout the study, 81% of patients that were hospitalized at baseline were discharged at endpoint, and 2% of patients hospitalized at baseline and subsequently discharged were readmitted before the end of the study. The rehospitalization rate in the case-control study conducted by Barrio et al. [37] was 19% in the LAI group and 42% in the OAP group (p = 0.136). In the post-hoc analyses by Dubois et al. [39], a reduction of -0.61 (1.24–0.63) in the TIMORES trial and 0.61 (0.87–0.26) in the e-STAR trial was found for the average number of hospitals stays per patient throughout the trial.

**3.2.5. Discontinuation.** In the study by Weiden et al. [50], 15 out of 18 (83.3%) of the OAP group and 15 out of 19 (78.9%) of patients discontinued their medication before 52 weeks. The 12-month RCT comparing LAI and oral risperidone conducted by Subotnik et al. [49] had an all-cause discontinuation rate of 25% in the LAI group and 37.2% in the OAP group, and significantly less patients in the LAI group discontinued due to inadequate response to treatment (p = 0.01). On the other hand, there was a greater all-cause discontinuation rate in the LAI group compared to the OAP group (62% versus 57%) in the 24-month open-label RCT conducted by Malla et al. [44]. The post-hoc analysis of a 12-month RCT conducted by Macfadden et al. [43] found a lower discontinuation rate among those recently diagnosed with schizophrenia (38.6%) compared to chronic patients (50.7%).

In the e-STAR trial, there was a 43% 2-year discontinuation rate whereas the TIMORES trial had a 1-year discontinuation rate of 16% [39]. The post-hoc analysis conducted by Emsley et al. [58] demonstrated that patients treated with RLAI had a lower all-cause discontinuation rate (26%) compared to those treated with OAPs (70.2%) within a 24 month period (p<0.005). The 6 month open-label studies by Napryeyenko et al. [45] and Parellada et al. [46] had an all-cause discontinuation rate of 14% and 27%, respectively.

**3.2.6 Remission.** In the post-hoc analysis conducted by Emsley et al. [40], a higher remission rate was achieved in the RLAI compared to OAP group (64% versus 40.4%; p = 0.028), while the study by Lasser et al. [42] found that 28% of patients not meeting criteria at baseline achieved remission for 6 months or greater during the 50-week trial period. Symptom based remission criteria was met by 63% of participants taking LAIs and 38% of participants taking OAPs in the 24-month case-control study by Barrio et al. comparing RLAI and various formulations of OAPs (p = 0.066) [37]. In the e-STAR and TIMORES trial, full remission was achieved by 20.2% and 34.0% of patients, respectively [39].

In the 6-month trial evaluated in the studies by both Napryeyenko et al. [45] and Rabinowitz et al. [47], PANSS symptom remission criteria was met by 32.3% of patients who entered the study. Of the patients who did not meet criteria at baseline, 40% attained and maintained remission status for at least 3 months [45]. The PAS highest level of functioning scale was found to be strongly associated with remission, with more patients in the "excellent" and "good" groups meeting and maintaining remission status [47].

**3.2.7. Adverse events.** In the open-label RCT conducted by Malla et al. [44], the most common adverse events (AEs) reported were insomnia (29%), anxiety (21%), headache (21%), and hyperkinesia (20%). The discontinuation rate due to AEs was 11.9% in patients taking LAI compared to 2.9% in patients taking OAP. In terms of self-reported AEs in the open-label RCT by Weiden et al. [50], patients in the LAI group reported weight gain (47.4%), anticholinergic issues (15.8%), sexual difficulties (11.8%), and 10.5% reported an extrapyramidal symptom (EPS)-related event. In the open-label RCT conducted by Subotnik et al. [49], 10% of patients taking LAI discontinued treatment due to AEs compared to 21% in the OAP group, but this difference was not statistically significant (p = 0.14). The mean BMI increased from

28.8 (5.1) kg/m$^2$ at baseline to 30.8 (6.1) kg/m$^2$ at end point. There were no significant differences found in terms of weight gain, BMI increase, prolactin levels, total cholesterol levels, or akathisia between the RLAI and OAP groups.

In the post-hoc analysis by Emsley et al. [58], RLAI treated patients had fewer extrapyramidal symptoms compared to those treated with oral haloperidol and oral risperidone based on the extrapyramidal symptom rating scale (ESRS) ($p \leq 0.001$). In the open-label study by Napryeyenko et al. [45], EPS-related AEs were experienced by 5.6% of patients, with the most common being akathisia (4.6%), extrapyramidal disorder (3.0%), and parkinsonism (2.6%). Weight gain was experienced by 3% of patients, and the mean BMI gain was 0.4 (1.35) kg/m$^2$ and mean weight change was 1.2 (3.8) kg. Additional AEs experienced included depression (2.3%), insomnia (2.3%), and anxiety (2.3%).

In the open-label study by Parellada et al. [46], 6% of patients discontinued due to AEs and treatment-emergent adverse events (TEAEs) were reported by 57% of patients. The most common TEAEs reported were insomnia (7%), depression (5%), anxiety (5%), and weight gain (4%). The mean weight increase was 1.8 kg and mean BMI increase was 0.6 kg/m$^2$. In the study conducted by Ruan et al. [48], the most common AEs included depression (12.9%), anxiety (9.7%), headache (9.7%), insomnia (6.4%), akathisia (3.2%), and non-acute dystonia (6.4%). In contrast to other studies finding weight gain among patients, the mean body weight decreased by 4.5 kg and mean BMI decreased by 4.2%.

The post-hoc analysis conducted by Macfadden et al. [43] found that 86.0% and 89.9% of patients experienced at least one AE of any type in the recent diagnosed group and chronic group, respectively. Seven percent of patients in the recent diagnosed group and 5.6% in the chronic group discontinued the study due to AEs. The most common AEs in the recent diagnosed and chronic diagnosed group were insomnia (31.6% and 26.7%, respectively), headache (15.8% and 19.2%), and anxiety (12.3% and 17.3%). The recent diagnosed group experienced a significant mean weight increase from baseline to LOCF end point (2.9 kg (7.5); p = 0.007), while the chronic group did not (0.5 kg (8.8) p = 0.344). Patients in the recent diagnosed group experienced a significantly increase in mean (SE) prolactin levels from baseline compared to the chronic group at LOCF end point (24.9 (4.7) versus 13.21 (2.1); p = 0.024).

**3.2.8. Miscellaneous.** In the RCT conducted by Bartzokis et al. [38], there was a significant decrease in frontal lobe white matter in the OAP group (-0.567 (0.593); p = 0.01) while there were no significant changes in the LAI group (0.164 (0.671); p = 0.47). The OAP group also experienced a significant increase in frontal gray matter volume (0.527 (0.814), p = 0.04) while the LAI group did not (-0.280 (0.976); p = 0.30). It was found that increased white matter volume was associated with faster reaction times on tasks that involved working memory (2-back task, p = 0.045) and mental flexibility (set shifting task, p = 0.029).

## 3.3 Paliperidone Palmitate (PP)

**3.3.1 Efficacy.** The open-label RCT by Schreiner et al. [32] observed a trend favouring PP over OAPs in terms of improvements in total PANSS score (mean change in total PANSS: -16.6 versus -14.1; p = 0.075). Additionally, 75.6% of patients taking PP achieved a $\geq 30\%$ change in PANSS compared to 69.4% of those taking OAP. The multi-phase study conducted by Bossie et al. [29] found that recent-onset patients had a greater mean change in total PANSS score compared to chronic patients (-27.3 (18.27); 31.4% reduction versus -22.2 (18.12); 27.0% reduction; p<0.001). The open-label study conducted by Zhang et al. [36] found significant improvements in terms of mean total PANSS scores in patients that switched to PP from their existing OAP treatment (-11.3 (21.38);17.6% reduction; p<0.0001). Additionally, the improvements were significantly greater in patients with a greater disease severity at

baseline (PANSS total score ≥ 70). Similarly, the study conducted by Cervone et al. [27] found that patients taking PP showed a reduction of psychotic symptoms and improved quality of life.

A case report investigated the effects of switching to PP from current oral medication in two clinical cases [30]. One patient (Subject A) was diagnosed with undifferentiated schizophrenia (DSM-IV) and mild intellectual disability and began treatment on aripiprazole 5 mg/day then switched to LAI formulation due to concerns about compliance. Prior to beginning PP treatment, the patient was given oral paliperidone 6 mg/day. Oral paliperidone improved his symptoms and PP treatment was subsequently started. At admission, his PANSS total score was 94 and improved to 66 at discharge. After 1 year, the patient was still receiving PP 50 mg every 28 days with no AEs and showed improvements in functioning.

**3.3.2 Adherence.** In the retrospective study conducted by Titus-Lay et al. [34], adherence was defined as the proportion of days with medication across the study period and was compared between OAP, LAI (only PP), and combined OAP/LAI treatment groups. The average proportion of days with medication was significantly different between the three groups (p<0.001): 32% (interquartile range: 41.7) in the OAP group, 76% (interquartile range: 63.9) in the LAI (PP) group, and 75% (interquartile range: 41.7) in the combined OAP/LAI group. A post-hoc comparison between OAP and LAI (PP) groups found statistical significance for adherence favoring LAI (PP) therapy (p = 0.008).

**3.3.3 Relapse.** In the multi-phase study conducted by Bossie et al. [29], the relapse rate was higher in the placebo treated group compared to the PP treated group (30% versus 10.2%; p = 0.029). Additionally, time to relapse was longer in the PP treatment group compared to the placebo group (p = 0.014). In the study conducted by Cervone et al. [27], only one patient experienced a symptom relapse and hospitalization was avoided by increasing the dosage of PP and the number of follow-up visits.

**3.3.4 Rehospitalization.** In the open-label RCT conducted by Alphs et al. [28], 2.4% and 5.7% of patients were rehospitalized in the PP and OAP group, respectively. In the 6-month naturalistic study conducted by Privat et al. [31], a significantly lower number of hospital readmissions were found in the LAI versus OAP treatment groups (0.73 (1.191) versus 2.41 (1.432) p = 0.000). Zhang et al. [36] conducted an open-label trial with recent-onset schizophrenia patients that switched from OAP to PP LAI, and found that while taking LAIs, 46 patients (8.8%) were hospitalized for psychiatric reasons. The mean number of hospitalizations in this study was 1.2 (1.0–1.3) and mean length of hospitalization stay being 36.6 (22.3–50.9) days while on LAIs [37].

**3.3.5 Discontinuation.** The all-cause discontinuation rate was 47.6% in the PP LAI group compared to 54.3% in the OAP group in the 15-month OL RCT conducted by Alphs et al. [28], with one patient in the OAP group discontinuing treatment due to inefficacy compared to 0 patients in the PP treated group. In an open-label trial comparing the effectiveness of PP LAI in recent-onset versus chronic schizoaffective disorder conducted by Bossie et al. [29], the discontinuation rate was 42.2% for recent-onset patients and 53.4% for chronic patients during the 13-week OL acute treatment and 12-week stabilization phases of the study. In the retrospective study evaluating the effectiveness of PP LAI in first-episode psychosis patients conducted by Cervone et al. [27], the discontinuation rate was 28.6% during the 6 month period of the study. In the post-hoc analysis of a multiphase trial conducted by Sliwa et al. [33], the discontinuation rate for recent-onset and chronic patients was 24.1% and 27% during the 9-week transition phase, respectively. During the 24-week maintenance phase, 35.4% of recent-onset and 43.8% of chronic patients discontinued and of the recent-onset patients that entered the DB relapse prevention phase, 6.8% discontinued compared to 18.8% of the chronic patients that entered this phase.

**3.3.6 Remission.** In the multi-phase open-label trial conducted by Bossie et al. [29], a greater number of recent-onset patients compared to chronic patients met stabilization criteria at the end of the 12-week OL stabilization phase (70.4% versus 60%; p = 0.010). In this study, stabilization was defined as a PANSS total score lower or equal to 70 and Young Mania Rating Scale (YMRS) and Hamilton Rating Scale for Depression 21-item (HAM-D-21) scores lower or equal to 12.

**3.3.7 Adverse events.** In the study conducted by Alphs et al. [28], 4.8% of patients in the LAI group 2.9% in the OAP group discontinued the study due to AEs. In the study conducted by Bossie et al. [29], 5.8% of patients discontinued the study due to AEs such as weight gain (9.7%), akasthisia (7.3%), and tremor (4.4%). Of the 521 patients included in the safety analysis in Zhang et al.'s [36] study, 82.3% experienced one or more TEAE during treatment with PP and 12.7% of patients discontinued due to TEAEs. Injection site pain was reported by 18.6% of patients, insomnia by 15.2%, akathisia by 13.4% and headache by 11.3%. However, the number of patients that reported injection site pain decreased by 15.8% from the first week of treatment to the end of the first month of treatment.

The post-hoc analysis conducted by Sliwa et al. [33] found that 31.5% of recent-onset and 42.7% of chronic patients reported an AE during the first month following initiation of PP treatment. Overall, recent-onset patients were less likely to experience AEs when compared to chronic patients. 9.3% of recent-onset versus 12.6% of chronic patients experienced an extra-pyramidal AE with non-specific extrapyramidal disorder being more common in recent-onset than chronic patients (4.6% versus 2.3%) and akathisia being more common in chronic patients (3.3% versus 1.9%). The average weight change at endpoint was 2.6 (± 0.9) kg in the recent onset group compared to 3.4 (± 0.7) kg in the chronic group (p = 0.42). Glucose-related AEs ocurred more in chronic versus recent-onset patients (5.1% versus 2.8%). On the other hand, prolactin related AEs were more common in recent-onset patients compared to chronically ill patients (7.9% versus 3.5%).

The case report conducted by Fàbrega et al. [30] included a patient who experienced multiple AEs (Subject B). This patient was a male diagnosed with psychotic disorder NOS and conduct disorder and LAI was started due to lack of insight. The patient started on oral paliperidone 3 mg/day prior to PP administration. Unfortunately, throughout the study, the patient experienced multiple readmissions and was nonadherent to the treatment. After 3 days of PP treatment, Subject B experienced an oculogyric crisis which was treated with biperiden 4 mg. As a maintenance treatment, PP 50 mg every 28 days with biperiden 4 mg/day was established. The patient experienced diurnal somnolesnce and concentration difficulties throughout the next 2 months and became non-compliant to treatment which led to behavioural difficulties and hospital readmission.

A case study conducted on a 34-year-old man that experienced his first episode of psychosis 2 years ago and diagnosed with paranoid schizophrenia found that treatment with PP 100 mg/month led to obsessive-compulsive symptoms [35]. These symptoms occurred after 6 weeks of treatment with PP and reducing the dosage of PP to 75 mg/month along with administering sertraline 200 mg/day led to remission of his obsessive-compulsive symptoms.

## 3.4 First-generation LAIs

**3.4.1 Fluphenazine decanoate.** In the RCT conducted by Kane et al. [52], 41% of patients in the placebo group relapsed compared to 0 patients in the group taking OAP or LAI fluphenazine. In the RCT conducted by Rifkin et al. [53], 68.4% of patients in the placebo group relapsed compared to 5.3% and 8.3% of those taking fluphenazine LAI and OAP, respectively. In the LAI group, 47.4% of patients discontinued the study compared to 16.7% in the OAP

group and 89.5% in the placebo group within the 12-month study period. Of these discontinuations, 26.3% were due to AEs in the LAI group compared to 4.2% in the OAP group.

The study conducted by Yee et al. [54] investigated P50 suppression ratios in patients taking LAI fluphenazine compared to oral risperidone. It has been shown that patients with schizophrenia often do not exhibit suppression to the second click in the auditory P50 component of the event-related potential, which is suggestive of deficits in attention and filtering of sensory information [59, 60]. In this study, P50 suppression was found to be impaired in recent-onset schizophrenia patients compared to control patients (p<0.05). It was found that there were no significant differences in P50 suppression ratios in patients taking fluphenazine decanoate compared to those taking risperidone. However, risperidone was superior to fluphenazine decanoate in terms of inhibition of P50 to the second click.

**3.4.2 Flupentixol decanoate.** In the prospective study conducted by Chiliza et al. [55] a PANSS reduction of 43.6% was found in patients taking Flupentixol LAI and 82% of patients experienced $\geq$ 50% improvement in their total PANSS score. The study conducted by Olivier et al. [56] recruited patients from the same catchment area as Chiliza et al. [55], but excluded patients that had an educational level lower than grade 7 and those who were not fluent in English or Afrikaans, as these were the languages used to conduct the MATRICS Consensus Cognitive Battery (MCCB), an assessment used to evaluate cognition in individuals with schizophrenia. In this study, there was also a total PANSS reduction of 43.6% (mean change: -41 (21.6)). In the study by Chiliza et al. [55], full remission (6-month maintenance) was achieved by 60% of patients within the 12-month study period. Of the patients that experienced a treatment response, 19% relapsed [56]. Furthermore, 23% of patients were hospitalized at baseline and 4% were rehospitalized throughout the study. The discontinuation rate was 28% in both the study by Chiliza et al. [55] and Olivier et al. [56], with both having a follow-up period of 12 months. The AEs reported in the study by Chiliza et al. [55] included akathisia (13%), parkinsonism (14%), stiffness (9%), tremor (11%), dystonia (10%), and dyskinesia (10.5%). Significant increases in body weight were found, with 56% of patients experiencing a weight gain of 7% or more. Other common AEs included depression (33%), excitement (13%) and anxiety (10%).

Furthermore, it was found that after 6 months of treatment, all patients demonstrated improvements in all cognitive domains, with no further improvements being observed after 12 months [56]. The mean MCCB composite score improved from 12 at baseline (95% CI: 9–15) to 25 at 6 months (95% CI: 22–28); p<0.0001. Higher PANSS total scores (p = 0.001) and higher baseline MCCB composite score (p = 0.01) were found to significantly predict cognitive improvements.

## 3.5 Aripiprazole LAI

Recently, an exploratory study was conducted on aripiprazole LAI treatment in first-episode schizophrenia patients [51]. In this study, patients were split up into an early age-at-onset group (18–21 years) and older age-at-onset group (22–26 years). In terms of efficacy, a 30.9% reduction (119.37 to 82.5) in mean total PANSS score was observed from baseline to 12 months. At the end of the study, 66% of patients taking aripiprazole LAI had a total PANSS score less than 80.

Throughout the 12-month duration of the study, 78% of patients remained adherent to their treatment. Of the 11 patients that discontinued the study, 7 were withdrawn from the study by the physician due to inefficacy or side effects, 2 switched to another LAI, 1 patient moved, and 1 dropped out due to intolerable akathisia. Other than the patient that withdrew

from the study due to akathisia, no other notable AEs were observed throughout the study period.

### 3.6 Unspecified LAIs

In the naturalistic study by Abdel-Baki et al. [25], the type of LAI used was not specified. In this study, 13.1% of patients in the sample received LAIs as a first-line treatment, and 54.8% of these patients continued taking LAIs throughout the entire 3-year duration of the study. Similarly, of the 86.9% of patients taking OAP as a first-line treatment, 58.7% of them remained on OAPs for the entire study. In this study, 70.6% of the patients in the LAI-only group versus 65.3% in the OAP-only group experienced at least one psychotic relapse throughout the study period. The time-to-first relapse was longer in the LAI-only group (703 days) compared to the OAP-only group (526 days). Additionally, 38.9% of patients taking OAPs only versus 52.0% taking LAIs only were hospitalized, with the mean time-to-first rehospitalization being longer in the LAI-only group (825 days) compared to the OAP-only group (772 days).

The 24-month case-control study by Morken et al. [26] also did not specify the type of LAI used. In this study, 12 patients were treated with LAIs throughout the study and among these patients, 6 had good adherence to treatment throughout the study. Of the adherent patients, 3 (50%) experienced a relapse compared to 5 patients in the non-adherent group (83.3%). Similarly, all 6 patients in the non-adherent LAI group were rehospitalized compared to only one in the adherent group, and the median days in hospital was 12 (range: 0–264). Patients taking LAIs had less improvements in Global Assessment of Functioning (GAF) score (0 (-18-15) (p = 0.040)) compared to those not taking LAIs. The GAF is used as a measure of the severity of an individual's mental illness based on their psychological, social, and occupational functioning [61]. Based on these findings, the authors concluded that LAI users tend to have a more severe illness and are less cooperative to treatment compared to non-LAI users.

### 3.7 Comparison between LAIs

In the nationwide prospective study conducted by Taipale et al. [57], the effectiveness of different LAIs was investigated including risperidone, fluphenazine, flupentixol, perphenazine, paliperidone, olanzapine, aripiprazole, haloperidol, and zuclopenthixol. Throughout the follow-up period, 57.9% of first-episode patients experienced a readmission to psychiatric inpatient care and the rate of all-cause hospitalization was 80%. The risk of psychiatric hospitalization was found to be lower in patients taking LAIs (first generation LAIs: HR: 0.46, 95% CI: 0.40–0.54; second generation LAIs: 0.45, 0.39–0.52) compared to OAPs (first generation OAP: 0.67, 0.60–0.74; second-generation OAP: 0.57, 0.53–0.61). In terms of specific drugs, the lowest risk of psychiatric rehospitalization among patients with first-episode psychosis was found for flupentixol LAI (0.24, 0.12–0.49), olanzapine LAI (0.26, 0.16–0.44), and perphenazine LAI (0.39, 0.31–0.50) compared to no use of antipsychotics. The greatest risk of rehospitalization associated with LAIs included haloperidol (0.69, 0.48–1.0), aripiprazole (0.63, 0.3–1.34), and risperidone (0.48, 0.42–0.56).

## 4. Discussion

This comprehensive systematic review supplements a previous meta-analysis conducted by the study author [62] which demonstrated that LAIs are superior to OAPs in terms of reducing relapse and hospitalization rates in early psychosis patients. The present review included studies that evaluated the usage of first-generation and second-generation LAI antipsychotics in patients with schizophrenia with a duration of illness < 5 years. We included studies with a wide range of design including 8 RCTs [28, 32, 38, 44, 49, 50, 52, 53], 2 case reports [30, 35]

and non-RCT studies including open-label studies, exploratory studies, post-hoc analyses, case-control, and cohort studies. The inclusion of non-RCT, naturalistic study designs allow for a comprehensive overview of the overall effectiveness of LAIs in first-episode psychosis patients. Although non-RCT lack randomisation and are therefore more prone to selection bias, these studies provide valuable information on the efficacy of LAIs in a more naturalistic, real-world setting compared to RCTs [63]. Overall, it appears that LAIs are effective pharmacological treatments and may reduce the risk of relapse in early psychosis patients. Furthermore, patients with a more recent diagnosis of schizophrenia seem to respond better to treatment compared to patients with a longer duration of illness [29, 33, 43]. Common AEs observed in studies examining LAIs include EPS-related AEs, insomnia, weight gain, depression, anxiety, and headache.

Previous reviews have demonstrated the second-generation LAIs are superior to OAPs as they can improve symptoms [21], increase medication adherence [20], remission rates [18], reduce relapse rates [20, 21] and reduce hospitalization rates [20]. These findings are in line with the studies included in the present systematic review [28, 31, 34, 36]. However, these reviews mainly included studies that evaluated RLAIs, which highlights the need for additional studies that evaluate different LAI formulations. This need is further demonstrated a nationwide prospective study included in the present study that compared different LAI formulations [57]. The study authors found that only risperidone, perphenazine, olanzapine, and haloperidol LAIs were superior to the equivalent oral formulations in terms of risk of psychiatric rehospitalization [57]. Based on these findings, the authors concluded overall effectiveness of LAIs compared to OAPs may not be due to improved adherence and regular contact with health care professionals, but rather other pharmacological properties of different LAIs.

In a review of the current guidelines and clinical trial data for LAI treatment in first-episode schizophrenia patients, it was recommended that guidelines are updated to include LAIs as a treatment option for patients that are nonadherent to medication, regardless of the stage of their illness [19]. As LAI users may have more severe illness and are less cooperative to treatment compared to non-LAI users [26], they may particularly benefit from the use of LAIs given their known efficacy in preventing relapse caused by non-compliance to treatment [19]. However, there are negatives to using LAIs such as include not being able to quickly adjust the dose of LAI formulation [19]. Furthermore, the use of LAIs may cause patients to become discouraged or unmotivated to recover as LAIs are often perceived as a treatment for those with more severe illnesses [19]. Therefore, it is important to take a person-centred approach and educate patients about the benefits of using LAIs over OAPs [15].

## 4.1 Limitations

It is important to note that among the studies included in this study, the criteria used for early and recent-onset schizophrenia ranged from a duration of illness of $\leq$ 2 years to $\leq$ 5 years. It is important to establish a consistent definition of early and recent-onset schizophrenia to allow for better evaluation of the effectiveness of LAIs in this specific patient population. Also, the different criteria and definitions used for remission and relapse among the included studies in this review makes it difficult to draw definite conclusions on the effects of LAIs on these two outcomes. Our included studies originated from various countries, raising concern regarding differences in practice patterns and guidelines between countries. Furthermore, the majority of studies included in the present analysis involved the use of RLAI or PP. Therefore, it is important that additional studies are conducted on the effectiveness of a variety of different LAI formulations such as the cohort study conducted by Taipale et al. [57]. Finally, the characterization of drug-related side-effects in many studies was not exhaustive and differed between

reports. The second-generation antipsychotic drugs are widely recognized to have pronounced cardiometabolic side-effects [64–66], and thus more detailed comparisons between groups other than simple weight gain would have been informative.

## 5. Conclusion

Overall, the present systematic review provides a comprehensive update on LAIs in early psychosis patients, covering clinical efficacy, rehospitalization and relapse, medication adherence and side-effects. The present findings aligns with the literature, which indicates that that LAIs are a relatively safe and effective treatment for early psychosis patients, although the majority of evidence is based on RLAI and PP. Clinical benefits may occur when health care providers are able to focus on providing patient-centered care and discuss the option of LAI treatment with early psychosis patients [16]. Educating patients and families on the benefits of LAIs may help them choose the best treatment option and achieve optimal clinical outcomes. However, there is still a critical need for high quality RCTs that compare the effectiveness and safety of LAIs compared to OAPs in the early stages of schizophrenia.

## Supporting information

**S1 Checklist. PRISMA 2020 checklist.**
(DOCX)

## Author Contributions

**Conceptualization:** Lulu Lian, Alasdair M. Barr.

**Data curation:** Lulu Lian, David D. Kim, Diana Cázares.

**Formal analysis:** Lulu Lian, David D. Kim.

**Funding acquisition:** Ric M. Procyshyn, William G. Honer, Alasdair M. Barr.

**Investigation:** Lulu Lian, Diana Cázares.

**Project administration:** Alasdair M. Barr.

**Resources:** William G. Honer, Alasdair M. Barr.

**Supervision:** Ric M. Procyshyn, William G. Honer, Alasdair M. Barr.

**Validation:** David D. Kim.

**Writing – original draft:** Lulu Lian, Alasdair M. Barr.

**Writing – review & editing:** Lulu Lian, David D. Kim, Ric M. Procyshyn, Diana Cázares, William G. Honer, Alasdair M. Barr.

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
