## [Decision Letter · Decision Letter 0]

6 Oct 2021

PONE-D-21-23982Long-acting injectable antipsychotic drugs for early psychosis : A comprehensive systematic reviewPLOS ONE

Dear Dr. Barr,

Thank you for submitting your manuscript to PLOS ONE. After careful consideration, we feel that it has merit but does not fully meet PLOS ONE’s publication criteria as it currently stands. Therefore, we invite you to submit a revised version of the manuscript that addresses the points raised during the review process.

ACADEMIC EDITOR: Please see my comment and reviewer comments below. Please follow re-submission instructions carefully. Thanks.==============================

We look forward to receiving your revised manuscript.

Kind regards,

Kyle J Burghardt

Academic Editor

PLOS ONE

Journal Requirements:

"WGH has received consulting fees or sat on paid advisory boards for the Canadian Agency for Drugs and Technology in Health, AlphaSights, In Silico (unpaid), Newron, Translational Life Sciences and Otsuka/Lundbeck. RMP has received consulting fees or sat on paid advisory boards for Janssen, Lundbeck and Otsuka; is on the  speaker’s bureau for Janssen,

Lundbeck and Otsuka. All other authors have no conflict of interest to declare"

We note that you received funding from a commercial source: Lundbeck and Otsuka

Please include your amended Competing Interests Statement within your cover letter. We will change the online submission form on your behalf

3. We note that this manuscript is a systematic review or meta-analysis; our author guidelines therefore require that you use PRISMA guidance to help improve reporting quality of this type of study. Please upload copies of the completed PRISMA checklist as Supporting Information with a file name “PRISMA checklist

Additional Editor Comments (if provided):

Thank you for your patience in reviewing your article. Please see and response to reviewer reports attached. Please revise you manuscript according to these comments and provide a reviewer response table as instructed by PLOS.

Can you please double check reference #34. From my reading, there were 3 groups compared in this study, more LAIs other than paliperidone were included, and the adherence numbers of (25% versus 8.6% adherent; p<0.001) were not listed in #34 reference? Could you please re-check the accuracy of data reported for this reference and others within your report as a double check? Thanks.

Reviewers' comments:

Reviewer's Responses to Questions

**Comments to the Author**

1. Is the manuscript technically sound, and do the data support the conclusions?

Reviewer #1: Partly

Reviewer #2: Yes

Reviewer #3: Yes

2. Has the statistical analysis been performed appropriately and rigorously? 

Reviewer #1: N/A

Reviewer #2: N/A

Reviewer #3: N/A

3. Have the authors made all data underlying the findings in their manuscript fully available?

Reviewer #1: Yes

Reviewer #2: Yes

Reviewer #3: Yes

4. Is the manuscript presented in an intelligible fashion and written in standard English?

Reviewer #1: Yes

Reviewer #2: Yes

Reviewer #3: Yes

5. Review Comments to the Author

Reviewer #1: Line 53: The word “however” doesn’t seem to fit after the previous sentence. Consider removing.

Line 55: Nonadherence rate of 39% seems too specific to one program. Consider finding a more generalized estimate among treatment programs with this same population.

Line 56: Consider the word “misuse” in place of “abuse”. This term is considered more stigmatizing and many substance use disorder organizations are urging healthcare professionals to switch the language to be less stigmatizing.

Line 67: Missing the word “to”.

Line 265: Comma missing after [49].

Line 300: Missing comma and a space after [38].

Line 359: Try to avoid starting a sentence with a number if possible.

Line 388: Try to avoid starting a sentence with a number if possible.

Line 396: A space is needed between “who” and “experienced”.

Line 516: Consider using person-first language such as “patients with schizophrenia”.

Reviewer #2: Overall this was a well written manuscript. The limitations that I marked while reading through the article were all pointed out in the limitation section of the manuscript. I believe the authors did touch on this, but when working with different definitions of remission, relapse, etc. it can be harder to generalize the results.

Some of the studies used DSMIV criteria for the diagnosis however DSM5 was released in 2013 I believe, so I wonder why some of the studies dating in 20019, 2018, still utilized DSMIV

It may have been beneficial to make one of the inclusion criteria "LAIs approved in each country". Because for some of the results Flupentixol (for example) had an advantage but since it is not available in the US, I am unable to utilize that information

Reviewer #3: Thank you for allowing me to review your work. The use of LAIs in early psychosis is an important topic and I believe your study will add to the growing body of evidence to support this practice. Please see below for specific comments:

• Introduction line 61: There have been studies of the economic impact of LAIs that you could cite here which help to address the questions readers may have about the increased cost of LAI formulations compared to OAPs.

• Introduction line 68: Consider elaborating more on why LAIs are useful in patients with poor adherence for those who may not be as familiar with the medications (i.e. they can be dosed less frequently than OAPs).

• Introduction lines 77-79: What is the benefit of including other study types and first-generation agents? (e.g. generalizability, sample size, real-world applicability, etc.)

• Methods line 83: State here that you included studies that did not compared efficacy of LAI to OAP- the reader may wonder this since it was the focus of your results reported.

• Table 1: Given the size of the table, consider writing out some lesser-known abbreviations (e.g. OL = open-label, NS = not specified, DB = double-blind?, DUP = duration untreated psychosis?) since there is space to do this. It saves the reader from having to scroll up and down several pages each time they need to look up an abbreviation.

• Table 1: The Dubois 2014 article describes e-STAR and TIMORES groups- these acronyms will need to be spelled out in the footnote and/or some indication given of how the 2 groups differed for the table entry to make sense to the reader.

• Table 1: Convert all study durations to months for consistency and ease of comparison.

• Table 1: Be consistent with how age is reported (e.g. sometimes it is median, sometimes range is included, Ruan 2010 has +/- for SD instead of (SD)) and if possible, calculate mean (SD) for all.

• Results: Include the follow up period for discontinuation and remission wherever possible.

• Results: Include measures of statistical significant wherever possible when comparing outcomes between LAI and OAP groups.

• Results line 150: Define LOCF- not sure what this endpoint is and if this study is comparing LAI vs. OAP like the others.

• Results line 156: This study needs to be explained more clearly- what made the e-STAR and TIMORES groups different that accounted for the change observed in CGI-S?

• Results line 159: Is this the LAI group specifically? How did it compare to the OAP group?

• Results lines 162-175: It would be helpful to remind the reader that these were studies with no comparator group, so you are only reporting the outcomes for the LAI (unlike the studies previously described in this section). The description of “open-label” is not sufficient for the reader to know this as many RCTs are also open label.

• Results lines 306-307: This statement is an interpretation of the results and should be in the discussion section.

• Results lines 336-337: Define IQR.

• Results lines 351-354: It may be helpful to mention again that these patients were switched from OAP to LAI PP and that these rates are on LAI.

• Results lines 360-361: Remind the reader that Bossie compares recent vs. chronic and Cervone is PP and has no comparison group. It is unclear as currently which groups you are referring to for the discontinuation rates reported.

• Results lines 405-407: These statements belong in the discussion section.

• Results line 439: Describe the MATRICS MCCB assessment.

• Results lines 462-465: Not sure how these results are relevant to your study aims. I would suggest leaving them out.

• Results line 489: Define GAF.

• Results lines 489-491: This statement belongs in the discussion section.

• Results lines 508-510: This statement belongs in the discussion section.

• Discussion: This section needs to be improved. Most of the discussion is devoted to the findings of other reviews. It appears that your main conclusion from your study is that “Overall, it seems that LAIs are effective pharmacological treatments and may reduce the risk of relapse in early psychosis patients.” This is something that your review can add support for but is rather weak based on your findings alone since there was no meta-analysis performed. It appears that your aim was to include a wider range of trial designs for a more “comprehensive” review on this topic, which I believe is a strength that makes your study unique. I would go on to explain more about why this is important and why we need more reviews that include naturalistic study designs to assess real-world utility of LAIs. However, you go on to conclude that there is a need for more RCTs- will these be as useful for demonstrating the benefit of LAIs? See the following studies:

o Haddad PM, Taylor M, Niaz OS. First-generation antipsychotic long-acting injections v. oral antipsychotics in schizophrenia: systematic review of randomised controlled trials and observational studies. Br J Psychiatry Suppl. 2009 Nov;52:S20-8. doi: 10.1192/bjp.195.52.s20. PMID: 19880913.

o Biagi E, Capuzzi E, Colmegna F, Mascarini A, Brambilla G, Ornaghi A, Santambrogio J, Clerici M. Long-Acting Injectable Antipsychotics in Schizophrenia: Literature Review and Practical Perspective, with a Focus on Aripiprazole Once-Monthly. Adv Ther. 2017 May;34(5):1036-1048. doi: 10.1007/s12325-017-0507-x. Epub 2017 Apr 5. PMID: 28382557; PMCID: PMC5427126.

• Limitations: Discuss the issues with including studies from different countries (i.e. practice patterns/guidelines, cost/insurance models, utilization rates, etc.).

6. PLOS authors have the option to publish the peer review history of their article (what does this mean?). If published, this will include your full peer review and any attached files.

Reviewer #1: No

Reviewer #2: No

Reviewer #3: **Yes: **Shaina Schwartz

---

## [Author Response · Author response to Decision Letter 0]

15 Mar 2022

We have uploaded a detailed file (Response to Reviewers) which describes in full how we have responded to each point.

---

## [Editor Report · Decision Letter 1]

18 Apr 2022

Long-acting injectable antipsychotics for early psychosis: A comprehensive systematic review

PONE-D-21-23982R1

Dear Dr. Barr,

We’re pleased to inform you that your manuscript has been judged scientifically suitable for publication and will be formally accepted for publication once it meets all outstanding technical requirements.

Kind regards,

Kyle J Burghardt

Academic Editor

PLOS ONE

Additional Editor Comments (optional):

Thank you for your thorough and thoughtful revisions based on reviewer comments.
---

## [Editor Report · Acceptance letter]

21 Apr 2022

PONE-D-21-23982R1 

Long-acting injectable antipsychotics for early psychosis: A comprehensive systematic review 

Dear Dr. Barr:

I'm pleased to inform you that your manuscript has been deemed suitable for publication in PLOS ONE. Congratulations! Your manuscript is now with our production department. 

Kind regards, 

on behalf of

Dr. Kyle J Burghardt 

Academic Editor

PLOS ONE